# *Pseudomonas aeruginosa* and the Complement System: A Review of the Evasion Strategies

**DOI:** 10.3390/microorganisms11030664

**Published:** 2023-03-06

**Authors:** Alex González-Alsina, Margalida Mateu-Borrás, Antonio Doménech-Sánchez, Sebastián Albertí

**Affiliations:** 1Instituto Universitario de Investigación en Ciencias de la Salud, Universidad de las Islas Baleares, 07122 Palma de Mallorca, Spain; 2Instituto de Investigación Sanitaria de les Illes Balears, 07122 Palma de Mallorca, Spain

**Keywords:** complement system, *Pseudomonas aeruginosa*, immune evasion, bloodstream infection

## Abstract

The increasing emergence of multidrug resistant isolates of *P. aeruginosa* causes major problems in hospitals worldwide. This concern is particularly significant in bloodstream infections that progress rapidly, with a high number of deaths within the first hours and without time to select the most appropriate treatment. In fact, despite improvements in antimicrobial therapy and hospital care, *P. aeruginosa* bacteremia remains fatal in about 30% of cases. The complement system is a main defensive mechanism in blood against this pathogen. This system can mark bacteria for phagocytosis or directly lyse it via the insertion of a membrane attack complex in the bacterial membrane. *P. aeruginosa* exploits different strategies to resist complement attack. In this review for the special issue on “bacterial pathogens associated with bacteriemia”, we present an overview of the interactions between *P. aeruginosa* and the complement components and strategies used by this pathogen to prevent recognition and killing by the complement system. A thorough understanding of these interactions will be critical in order to develop drugs to counteract bacterial evasion mechanisms.

## 1. Introduction

*Pseudomonas aeruginosa* is one of the most common Gram-negative organisms causing nosocomial bacteremia [1,2]. Despite improvements in antimicrobial therapy and hospital care, *P. aeruginosa* bacteremia remains fatal in about 30% of cases [3,4]. Risk factors for *P. aeruginosa* bloodstream infections are multifactorial and include the intrinsic virulence of the microorganism and some underlying host conditions [5,6,7]. Moreover, the appearance of resistant strains to multiple antibiotics is increasing [8,9]. Therefore, treating infections caused by *P. aeruginosa* is becoming increasingly challenging. This fact, in conjunction with the broad arsenal of virulence factors [10], has made the World Health Organization include *P. aeruginosa* in the list of most dangerous pathogens [11].

A remarkable feature of *P. aeruginosa* bloodstream infections is their rapid progression. Many deaths occur shortly after infection [8,12], without time to develop an effective adaptive immune response. Therefore, it is up to the innate immune response to counteract the infection. This observation suggests that the complement system, the main early innate immune effector in the blood, plays a critical role in this type of infection.

This review will provide a comprehensive overview and update on the interactions of the complement system with *P. aeruginosa* and their biological significance. After a short introduction of the complement system, we will focus on the *P. aeruginosa* components that mediate the interaction with the complement components and the evasion strategies exploited by this pathogen to elude the complement system effects.

## 2. The Complement System

The complement system is an important effector arm of the immune system and, together with the phagocytes, is the main system responsible for innate immunity. The literature on the complement system is overly abundant, and the reader is referred to reviews [13,14,15] for additional knowledge on this subject. The complement system is integrated into many processes, but one of its main functions is the defense against infections. The complement system may directly lyse the bacterial cells, opsonize them to facilitate the recognition by the phagocytes, and release anaphylatoxins that promote the phagocyte recruitment and pro-inflammatory state at the site of infection.

The activation of the complement system occurs sequentially through the generation of active and unstable enzymes with serine protease activity. Depending on the activating molecule of the complement system, three pathways can be distinguished: the classical pathway (CP), the lectin pathway (LP), and the alternative pathway (AP). Activation by either of these results in the formation of the enzyme C3 convertase, a complex that acts on the central component of the complement system, the C3 molecule. A schematic representation of the complement system activation pathways is shown in Figure 1.

The first pathway to be described was the CP. The first component of this pathway is C1, a multimeric protein composed of C1q and the Ca^2+^ ion-dependent proenzymes C1r_2_ and C1s_2_. This pathway is activated by the binding of the C1q component to its ligands, mainly immunocomplexes of IgM and certain IgG isotypes, as well as pathogen-associated molecular patterns such as the lipid A or the bacterial porins. C-reactive protein (CRP) and molecules expressed on the surface of apoptotic cells are also C1q binding molecules. The binding of C1q to its ligand produces a conformational change that induces the autocatalytic activation of the serine protease C1r, which cleaves and activates the serine protease C1s. Subsequently, C1s hydrolyzes C4 to C4a and C4b. C4a is a small peptide of about 9 kDa, and C4b is a protein with a high molecular mass (190 kDa), which serves as a platform for the formation of the C3 convertase (C4b2a). C4b binds covalently through a thioester domain to amino and hydroxyl groups on the pathogen’s surface, where it recruits the C2 component to form C4bC2. The C1s then act on C2, generating C2a and C2b. Finally, C2a remains bound to C4b, giving rise to the enzyme C3 convertase of the CP (C4bC2a).

The LP is activated by the binding of mannose-binding lectin, a protein structurally similar to C1q, to certain glycoprotein and glycolipid carbohydrates present on the surfaces of microorganisms. More precisely, mannose-binding lectin recognizes the -OH terminal groups 3′ and 4′ from some sugars, such as glucose or mannose monosaccharides and N-acetyl glucosamine. Specifically, these sugars are rarely present on cell surfaces or host proteins, but often in bacterial, virus, or apoptotic cells. The interaction induces the activation in plasma of the mannose-binding lectin associated proteases MASP-1 and MASP-2. As occurs with the C1 complex of the CP, these proteases are dependent on Ca^2+^ and homologous in function to C1r and C1s, thus generating the C4bC2a or C3 convertase complex of the LP, which is identical to the C3 convertase of the CP.

Finally, the AP is constitutively activated at basal intensity due to spontaneous hydrolysis of the thioester bond of the C3 component to C3(H_2_O), also known as tick-over. This new molecule has the ability to bind to the protease Factor B. Subsequently, Factor B is hydrolyzed by the serin protease Factor D, generating the initial convertase C3(H_2_O)Bb, which is present in low amounts in the serum. Moreover, this complex produces a reduced number of C3b molecules which, in turn, can originate the C3 convertase C3bBb (C3 convertase of the AP). Through this process, a molecule stabilized by properdin is obtained that is much more active than the initial convertase, allowing a much higher production of C3b. Ultimately, it should be noted that the hydrolysis product of C3 by this convertase generates more C3b, which, together with that produced by the C3 convertases of the CP and the LP, establishes a feedback loop. Therefore, it ensures the amplification of the response, being fast and efficient once the AP of the complement system is activated. Additionally, the process of generating active C3 or C3b also produces C3a molecules, potent anaphylatoxins that induce chemotaxis, mast cell degranulation, and opsonization.

To this extent, the result of complement activation by any of the three pathways is the production of a C3 convertase, which could be C3bBb or C4bC2a. The C3 convertase generates C3b molecules that are deposited on surfaces acting as opsonins, helping phagocytes, and amplifying the complement activation. In addition, the C3b component can bind to the C3 convertases, giving rise to the C5 convertases C3bC2aC3b and C3bBbC3b. This new link changes the substrate specificity of the active centers of the molecules located in C2a and Bb, which becomes the C5 component.

Consequently, the newly generated enzyme complexes hydrolyze C5 molecules, producing C5a and C5b. C5a is a potent anaphylatoxin that is released in serum, while the C5b molecule activates a non-enzymatic assembly process of the final components of the complement system, C6 to C9. At this point, the C5b component is highly labile and rapidly deactivated unless it binds to C6, stabilizing its activity. The C5bC6 complex, initially bound to C3b, binds to C7 and becomes amphipathic. It is then transferred to the membrane, where once anchored to the lipid bilayer, it binds to C8, which undergoes a conformational change exposing a hydrophobic region that interacts with the plasma membrane.

Eventually, the last stage of the complement begins with the formation of the Membrane Attack Complex (MAC). MAC is the union and polymerization of C9 components, a perforin-like molecule, to the multimeric complexes C5b678 (C5b-C8). This generates a pore-tubular structure that allows the flow of water and loss of electrolytes, thus destabilizing the cellular osmotic stability and causing the lysis of susceptible cells, such as Gram-negative bacteria.

Due to the rapid and powerful effect of the complement system activation as the first line of defense against infections, as well as the multiple effector functions that it triggers, there must be mechanisms to control its function. A series of complement regulatory proteins maintain the system in homeostasis, preventing it from being harmful at a systemic level and ensuring that it acts efficiently and locally on foreign surfaces. There are both soluble and membrane-associated regulators, most of which are negative regulators whose goal is to limit the destructive effects of the complement on self-surfaces. Among the membrane-associated complement regulatory proteins, a pair of groups can be differentiated. On the one hand, there are the inhibitors acting on the C3b and C4b components to prevent the formation of C3 and C5 convertases or to promote their dissociation. These are the Complement Receptor 1 (CR1) or CD35, the Membrane Cofactor Proteins (MCP) or CD46, and the Decay Accelerating Factor (DAF), or CD55. In this sense, CR1 and MCP act as membrane cofactors for Factor I, a component with serine protease activity capable of degrading C3b and C4b. In addition, CR1 shares the decay activity with DAF, being able to degrade the inactive form of C3b (iC3b), generating C3 derivatives such as C3c and C3dg. On the other hand, the CD59 inhibitor acts on the active C8 and C9. Its function is to bind to the C5b-C8 and C5b-C9 complexes, thus limiting its insertion into the membrane and inhibiting the MAC formation. Furthermore, two plasma proteins also regulate the formation of the terminal C5b-9 complex: clusterin and vitronectin (also known as protein S). Specifically, these regulatory proteins bind to the C5b-C8 complexes, preventing their insertion into the membrane and the assembly of C9 components. Another plasmatic regulator is the C1 inhibitor (C1-INH), a serine protease that can prevent the activation of the CP and LP by inactivating C1r, C1s, MASP1, and MASP2. In addition to all the regulators already mentioned, it is important to highlight properdin, the only positive regulator. This protein stabilizes the AP C3 convertase, increasing its half-life and inhibiting Factor H-mediated C3b degradation. Finally, it is worth highlighting the role of the soluble regulatory proteins C4 Binding Protein (C4BP), Factor H, and Factor H-like protein 1 (FHL-1). C4BP is a chief regulator of the CP and LP, while Factor H and FHL-1 inhibit the activation of the AP. The main function of C4BP is binding to C4b molecules, preventing the formation of the C3 convertase C4bC2a and accelerating its decay process, causing the release and permanent inactivation of C2a. In addition, it serves as a cofactor in Factor I-mediated proteolysis of cell-bound and soluble C4b, as well as soluble C3b. The main regulatory function of Factor H is the inhibition of the AP C3 convertase. Factor H competes both in the plasma and on the cell surfaces with Factor B by binding to C3b, thus preventing the formation of AP pro-convertase. Moreover, it also accelerates the dissociation of C3 and C5 convertase from AP and acts as a cofactor for Factor I in the proteolysis of C3b like FHL-1.

## 3. The Complement System in *P. aeruginosa* Infections

Bacterial infections are a complex and dynamic interaction between a wide range of microorganism virulence factors and a series of defense mechanisms deployed by the host to prevent infections and eliminate the pathogen. To this extent, many in vivo experiments have demonstrated that complement is particularly critical in providing a protective immune response against *P. aeruginosa* infections. An early study conducted by Gross et al., demonstrated that mice depleted of complement by using cobra venom factor were more susceptible to *P. aeruginosa* pneumonia than untreated mice [16]. Similarly, cobra venom factor reduced the capacity of mice to clear *P. aeruginosa* after corneal infection [17].

The importance of the complement has also been illustrated using knockout mice deficient in specific complement components. For instance, C5-deficient mice showed defective lung clearance of *P. aeruginosa* due to a reduction of the number of phagocytes recruited to the lung [18,19]. In a study by Mueller et al. [20], C3, C4, or Factor B deficient mice were infected intranasally to investigate the activation pathways involved in defense against lung infections by *P. aeruginosa*. The results demonstrated that healthy control mice and C4-deficient mice had much lower mortality rates and bacterial loads in the lungs and bloodstream than C3 and Factor B-deficient mice. Moreover, bacteria opsonized with serum from C3-deficient mice were less efficiently phagocytosed by neutrophils than those opsonized with serum from healthy mice. Taken together, these results indicate that the protection of the *P. aeruginosa*-infected mice relies mainly on the activation of the AP activation rather than on the CP.

Two separate studies have investigated the role of the LP to combat *P. aeruginosa* infections. In a first study, Møller-Kristensen et al., used a murine model of burn infection to demonstrate that LP plays a critical role in containing and preventing a systemic spread [21]. Conversely, Kenawy et al., observed that MASP-2-deficient mice did not display a higher mortality rate than healthy mice in a model of acute lung infection by *P. aeruginosa* [22]. It is likely that LP is critical to control bacteriemia but it is not important to contain pneumonia, suggesting that LP is not required unless the pathogen reaches the blood stream. Alternative explanations for these somehow contradictory results are the technical approaches used to investigate the role of the activation pathway (chemical inhibitors versus knockout mice), and the phenotypic features of the strain used in each study.

Finally, a recent study has demonstrated that, in human blood, complement is the main defensive mechanism against *P. aeruginosa*. This study found a highly variable survival rate in blood among different strains, independently of their origin and serotype. Furthermore, they observed that most *P. aeruginosa* clinical isolates form a subpopulation of transiently complement-tolerant bacterial cells, called “evaders”, that can persist within the bloodstream. These evaders may facilitate the persistence and dissemination of the microorganism [23].

Since activation of the complement system, in particular the AP, is one of the main strategies of the host to combat *P. aeruginosa* infections, several studies have tried to identify the bacterial target molecules of the central component C3.

A major C3-binding molecule is the major *P. aeruginosa* outer membrane protein, OprF. This porin allows the passage of specific ions such as Na^+^ and Cl^−^ and small charged molecules. Moreover, it is involved in the anaerobic respiration of bacteria and in the formation of biofilms [24]. Mishra et al., proved that an OprF-deficient *P. aeruginosa* mutant had lower C3b deposition than the parental strain. By binding less C3b, the OprF-deficient mutant was phagocytosed by neutrophils less efficiently than the parental strain. In addition, OprF expression significantly increased the C3b binding and killing of this microorganism in an *Escherichia coli* strain sensitive to the bactericidal effect of complement [25].

The outer membrane protein OprH is the most recent C3 acceptor component identified on the *P. aeruginosa* surface [26]. Immunoblot assays with purified human C3 and whole human serum proved that the 21 kDa OprH protein bound C3. Furthermore, a quantitative analysis revealed that both the parental strain and the OprH-deficient mutant supplemented with the *oprH* gene bound more C3 than an isogenic OprH-deficient mutant. Consequently, the parental strain and the complemented mutant were killed by opsonophagocytosis more efficiently than the OprH-deficient mutant by human neutrophils. However, OprH expression is regulated by the PhoQ/PhoP system, which responds to Mg^2+^ ion concentration. In the presence of blood concentrations of this ion, OprH is not expressed, suggesting that the OprH-C3-mediated opsonophagocytic killing is not biologically significant in *P. aeruginosa* bloodstream infections [26].

LPS from Gram-negative bacteria is a well-recognized complement-binding molecule. In this sense, a series of early studies demonstrated that *Salmonella spp* or *Klebsiella pneumoniae* LPS mediates the binding of C3 through the O antigen and C1q by the lipid A portion, playing a crucial role in complement consumption [27,28]. However, few studies have focused on the interaction of complement components with *P. aeruginosa* LPS [29]. Since LPS, more precisely, the core and the lipid A, are well-conserved structures among Gram-negative microorganisms, it is generally assumed that *P. aeruginosa* LPS binds complement activating components. In fact, it has been extensively shown that the LPS of this pathogen is a potent activator of the complement system [29,30,31].

## 4. Complement Evasion Strategies of *P. aeruginosa*

*P. aeruginosa*, like many other pathogens, has evolved different mechanisms and strategies to resist the complement system attack. These mechanisms may be classified into three types (Figure 2): (i) blockage of the binding of the complement activating components; (ii) the binding of inhibitory complement regulatory proteins; and (iii) the inactivation of complement components. In this section, we will review the *P. aeruginosa* virulence factors or structures involved in these strategies and the underlying mechanisms that contribute to complement resistance.

### 4.1. Blockage of the Binding of the Complement Activating Components

#### 4.1.1. Lipopolysaccharide O Antigen

LPS is the major component of the *P. aeruginosa* outer membrane and an essential element for the structural integrity of the bacterium. The structure of LPS is formed from the innermost to the outermost part by lipid A, the inner core, the outer core, and the O antigen or polysaccharide (Figure 3).

Lipid A is the innermost region of the LPS, and, due to its hydrophobic nature, it is anchored to the bacterium’s outer membrane. Structurally, it is formed by a basic structure that contains an N- and O-acylated diglucosamine bisphosphate backbone. All lipid A modifications lead to an altered host immune response [32,33,34]. Thus, it has been demonstrated that an increase in lipid A acetylation provides more resistance to antimicrobial peptides [35] and a more significant inflammatory response through the Toll-like receptor 4 (TLR-4) [36].

Attached to lipid A is the inner core, which contains two D-manno-oct-2-ulosonic acid residues and two L-glycero-D-manno-heptose residues [37,38], the latter attached to a 7-O-carbamyl group [39]. The inner core is highly conserved and is essential for cell viability. The outer core consists of groups of sugars that can be synthesized from two different glycoforms or isoforms, depending on the position of the residues they contain. Furthermore, the hydroxyl groups of sugars are usually O-acetylated [40].

Finally, the O antigen consists of repeating linear or ramified oligosaccharide units. *P. aeruginosa* produces two different O antigen forms: A-band (also known as common antigen), and B-band, which is the heteropolymer responsible for serogroup specificity. This repetitive glycan polymer is highly immunogenic and responsible for the LPS serotype. There are 20 serotypes made up of units of 3–4 monosaccharides, except for serotype O7, consisting of disaccharides [38,41]. Furthermore, strains from patients with chronic infections, such as CF patients, are generally unable to synthesize the O antigen, the so-called “rough” LPS. On the contrary, if the pathogen presents an O antigen, it would have the so-called “smooth” LPS.

The biosynthesis of the O antigen has been related to 11 different loci. Among them, it is worth highlighting the product of the *wzz1* and *wzz2* genes, responsible for adding saccharide units to the B band of the O antigen. In this sense, the Wzz-1 protein incorporates between 12 and 30 saccharide units generating an long O-side chain, while the protein Wzz-2 incorporates between 40 to 50 saccharide units, rendering a very long O-side chain [42].

Many studies support the significance of the O antigen in the resistance to the complement-mediated effects [30,43,44,45]. *P. aeruginosa* strains with a smooth LPS are resistant to serum killing and are the predominant phenotype in *P. aeruginosa* strains causing acute infections. On the contrary, strains with rough LPS are more frequent in chronic infections and are more sensitive to the bactericidal effect of the complement [46,47]. Thus, the serum sensitivity phenotype exhibited by the chronic infection respiratory isolates might explain why they are generally confined to the lung, where the levels of the terminal complement components are usually low, and are rarely spread to the bloodstream [48].

The number of O-side chain saccharide units required to confer resistance to the microorganism has not been determined. Early pioneering serum survival experiments performed by Kintz and Goldberg using isogenic strains deficient in *wzz1* or *wzz2* derived from *P. aeruginosa* PA103 demonstrated that the *wzz2*-deficient mutant was as resistant as the parental strain. In contrast, the *wzz1*-deficient mutant was more susceptible [42,43]. However, in preliminary experiments conducted in our laboratory with *P. aeruginosa* PAO1, the Wzz1-deficient mutant and the parent strain were serum-resistant, while the Wzz2-deficient mutant was susceptible to the bactericidal effect of the complement. It appears that the contribution of Wzz1 and Wzz2, which modulate the incorporation of saccharide units to the O side chain, on serum resistance may depend on the LPS O serotype and other strain-specific features that remain unknown.

In any case, the O antigen contributes to complement resistance by blocking the access of the activator components, mainly C3, to the cell surface targets.

#### 4.1.2. Capsule Polysaccharides

*P. aeruginosa* produces three exopolysaccharides: alginate, the main determinant of the mucoid appearance, Psl (polysaccharide synthesis locus), and Pel (pellicle formation). There is much evidence to suggest that alginate and PsI mediate complement resistance. However, the contribution of Pel to complement resistance remains poorly investigated.

Alginate is a partially acetylated linear polymer formed by β(1→4) bonds of D-mannuronic acid and L-glucuronic acid [49]. Alginate synthesis is encoded by the *algD-algA* operon, which comprises 12 genes (*algD, alg8, alg44, algK, algE, algG, algX, algL, algI, algJ, algF*, and *algA*), each of which is involved in some of the alginate’s processes of synthesis, modification or export [49]. The expression of the *algD-algA* operon is mainly regulated by the *algD* promoter. A key element in the regulation of alginate production is the sigma factor (σ22), also known as AlgU or AlgT. This element induces the expression of AlgD and increases the expression of AlgR or AlgB, proteins that improve *algD* transcription [49]. The σ22 factor is encoded by *algU*, which is part of the *algU-mucABCD* operon and is regulated by the *mucA* and *mucB* genes products. Specifically, MucA prevents the binding of AlgU to the promoter zone of the *algD-algA* operon, preventing alginate synthesis. Consequently, mutations in *mucA* or *mucB* genes in strains with a mucoid phenotype are typical, since their inactivation leads to AlgU deregulation and alginate overproduction due to the permanent activation of the algD-algA operon [50]. More than 80% of the mucoid strains isolated in CF patients have mutations in *mucA* [50,51].

The products of *algF, algI, algJ, algK, algL,* and *algX* genes participate in the modification of alginate, which results in the mannuronic acid acetylation of the hydroxyl groups [49], giving rise to the mature alginate, which is transported across the outer membrane by AlgE [49]. This acetylation is pivotal for the bacteria, since it avoids complement opsonization. Specifically, it prevents the hydroxyl groups from interacting with the C3b and C4b components and avoids the activation of the AP. In fact, it has been shown that the algJ mutant strains are more sensitive to the bactericidal effect of the complement since they present a lower percentage of acetylation [33].

Psl contributes to *P. aeruginosa’s* resistance against the complement system. Moreover, it is also involved in biofilm formation and regulates bacterial attachment to epithelial cells. Psl reduces the deposition of the C3b component on the surface of the microorganism, thereby hindering the process of opsonophagocytosis [52]. To evidence this, Mishra et al., determined the phagocytosis rate of a *P. aeruginosa* strains collection that produced different amounts of the Psl polysaccharide. They observed that those strains that expressed lower amounts of Psl deposited more C3, C5, and C7 and were phagocytosed more efficiently than those that expressed higher amounts of the polysaccharide. Thus, Psl-deficient strains exhibited an attenuated virulence in vivo [52]. In a later study, Pestrak et al., demonstrated that the absence of the Psl exopolysaccharide increased C3 deposition and neutrophil-mediated phagocytosis [53]. Consistent with these results, Jones et al., demonstrated that Psl and not alginate is the main inhibitor of C3 deposition on the surface of the microorganism [54]. They observed that an isogenic mucoid Psl-deficient mutant bound significantly more C3 than the parental mucoid strain or the isogenic Psl-deficient mutant complemented with the wild-type *psl* operon. Conversely, mutation of algD, which renders a non-mucoid strain, in a Psl-producing strain had no impact on the binding of C3 to *P. aeruginosa* [54]. Thus, Psl is a critical virulence factor of *P. aeruginosa* for the evasion of a complement lytic effect and opsonization since it prevents C3 deposition on the bacterial surface.

#### 4.1.3. Biofilm

Biofilm is a dense environment with a very complex organization composed of a bacterial community surrounded by a matrix of exopolysaccharides, DNA, and proteins, which can be formed on both biotic and abiotic surfaces [55,56,57]. The formation of biofilm has different stages. It begins when planktonic bacteria adhere to an inert surface, such as intravenous catheters or joint prostheses, or to in vivo surfaces such as teeth, heart valves, or even the lungs of patients with some pathologies such as CF or chronic obstructive pulmonary disease. Despite the fact that there is no evidence that *P. aeruginosa* forms biofilms during systemic infections or bacteriemia, biofilm formation in intravenous catheters is a risk factor of bloodstream infections [55]. Moreover, biofilms are a major concern in chronic airway respiratory infections [58]. Biofilms are formed in response to signals such as bacterial density, the availability of nutrients and energy sources, temperature, or osmolarity, among others. Once bacterial adhesion to the surface has occurred, the transcription of bacterial exopolysaccharides matrix *pel* and *psl* genes is activated. Subsequently, a bacterial multiplication takes place that will give rise to microcolonies that will remain embedded in this matrix, which also contains extracellular DNA, proteins, and lipids. In addition to the vital importance of the expression of specific genes, another significant factor for the complete formation of the biofilm is the correct functioning of the quorum sensing system [59]. Over time, the number of bacteria forming the biofilm will increase with its thickness and size. This is partly because of the overproduction of alginate, which accounts for 90% of the biofilm from the total biofilm organic matter, and gives the characteristic mucosal phenotype of most CF chronic isolates. Finally, when the biofilm is fully formed and has “matured”, bacteria will be released from the outermost part of it, colonizing other surfaces to generate new biofilms [60,61].

Biofilm protects bacteria against unfavorable conditions such as antimicrobial therapy and the host’s immune response, including oxidative stress, nutrient restriction, polymorphonuclear cell (PMN) infiltration, opsonization by complement or antibodies, and phagocytosis [59,62,63].

Despite the fact that there are numerous studies on the *P. aeruginosa*-complement interaction in planktonic growth, the role of the immune system during biofilm infections is not well established. Interestingly, most isolates from CF patients are sensitive to the complement lytic action, since they present an LPS devoid of O antigen, the main complement resistance factor described in this microorganism. Nonetheless, these microorganisms can continue to grow within the biofilms formed in the lungs of these patients. It is generally accepted that this is because biofilm protects the microorganisms against the action of the complement system. Nevertheless, the underlying molecular bases of the protection mechanisms are unknown. Some authors suggest that the biofilm structure itself confers resistance to complement. In this sense, Jensen et al., confirmed that *P. aeruginosa* biofilms activate complement less efficiently than bacteria in the planktonic state [29]. This observation is consistent with the fact that alginate and Psl exopolysaccharide, major components of biofilm, reduce complement activation and deposition on bacteria, as described in the previous section.

Along with these observations, it is worth adding the profound lack of knowledge about the state of the complement system in the lungs of patients with chronic respiratory infections caused by *P. aeruginosa*. Analysis of sputum from CF-infected patients suggests the presence of C3 and C5 in the lung. Indeed, elevated C5a levels, as a result of C5 activation by any of the complement system activation pathways, is a biomarker of poor disease prognosis [64]. However, there is no conclusive evidence that the complement components of the lytic pathway (C6, C7, C8, C9) are present in the lung, which could explain the presence of complement-sensitive bacteria in the lung.

Overall, although there is clear clinical evidence of the presence of bacteria sensitive to the complement lytic action in the lungs of CF patients, the mechanisms that permit resistance to this effect in the context of biofilm growth are still poorly understood.

#### 4.1.4. Cloaking Antibodies

Wells et al., have recently described a new strategy that *P. aeruginosa* exploits to evade the complement system by using specific antibodies that protect the bacterium instead of marking it for destruction [65]. They found that the serum from patients with bronchiectasis was unable to lyse the autologous infecting bacterial strain. However, all strains isolated from patients were susceptible to serum killing when they were exposed to sera from healthy individuals, which suggested the presence of some component, present in the serum of bronchiectasis patients, but absent in healthy individuals’ serum, that prevented the lysis of the bacteria. Hence, they determined that protection was mediated by high titers of IgG_2_ antibodies specific for the O antigen that cloak the surface of the bacteria, blocking access to the membrane. Cloaking antibodies have been found to be prevalent in CF patients. Pham et al., demonstrated that 32% of the CF patient sera contain antibodies that inhibit the complement-mediated killing of *P. aeruginosa* [66].

On the other hand, the presence of IgA antibodies can inhibit complement-mediated killing by providing a physical block to protect bacteria against host MAC activity. Thus, IgA, the main defense antibody of the respiratory epithelium, may be crucial in the context of pulmonary infection by *P. aeruginosa* [66].

### 4.2. Binding of Inhibitory Complement Regulatory Proteins

#### 4.2.1. Elongation Factor Tu (EF-Tu)

EF-Tu is a highly conserved 43 kDa cytoplasmic protein encoded by the genes *tufA* and *tufB*, which have identical sequences. It is one of the most abundant bacterial proteins, representing 6% of the total cell protein. Its canonical function is to transport aminoacylated tRNAs to the ribosome [67]. However, its presence on the bacterial surface has been described, where it can interact with complement components such as Factor H, FHL-1, and CFHR-1 [68]. The binding of Factor H promotes the degradation of C3b and blocks the activation of the AP [68].

#### 4.2.2. Dihydrolipoamide Dehydrogenase (Lpd)

Lpd is a cytoplasmic protein that can also be found on the bacterial surface. When found in the cytoplasm, it catalyzes the transfer of electrons between pyrimidine nucleotides and disulfide components, as it is part of the pyruvate dehydrogenase enzyme complex. However, when found on the bacterial surface, it can bind Factor H, FHL-1, and CFHR-1 [69]. The ability of *P. aeruginosa* to bind Factor H, FHL-1, and CFHR-1 facilitates the evasion of the complement system attack, and it is crucial for the survival of *P. aeruginosa,* as blocking this protein with a specific antiserum reduces bacterial survival in human serum [69]. Furthermore, Lpd binds vitronectin and clusterin, inhibiting the MAC formation on the bacterial surface [70]. This mechanism is of particular interest in respiratory infections, since it has been proved that during pneumonia caused by *P. aeruginosa*, the levels of vitronectin in the bronchoalveolar fluid increase [70,71].

#### 4.2.3. OprD

OprD is an outer membrane protein involved in the uptake of small peptides and positively charged amino acids such as lysine [72]. It is also permeable to carbapenems; therefore, the appearance of deficient mutants in this protein is common, especially in chronic respiratory infections. In fact, the mutation of this protein constitutes the most common mechanism of resistance to carbapenems [73]. Moreover, Paulsson et al., identified OprD as a vitronectin-binding protein on the surface of *P. aeruginosa.* They reported that airway isolates from CF patients present more robust vitronectin-binding phenotypes than bloodstream isolates. These results are reasonable because vitronectin is produced by pulmonary epithelial cells and is upregulated in the airway of CF patients, being available in the lower respiratory tract [74]. In this sense, it has been suggested that OprD participates in the evasion of the complement system by binding vitronectin and using it as a bridging molecule that contributes to epithelial adherence, at least during initial colonization and exacerbations [74].

### 4.3. Inactivation of Complement Components

#### 4.3.1. Exoproteases

Another strategy to evade the innate immune system is the inactivation of complement proteins by extracellular proteases. *P. aeruginosa* secretes several proteases with the capacity to inactivate or degrade components of the complement system including elastase B (LasB), alkaline protease A (AprA), Protease IV (PIV) and Pseudomonas Small Protease (PASP). The expression of these proteases is highly influenced by environmental factors such as the availability of nutrients, pH, and temperature, as well as the phase of the cell cycle or the type of growth (planktonic or biofilm). In general, its production is higher in acute infections such as bacteremia and acute pneumonia compared with chronic infections, suggesting a role in the initial colonization of tissues [75,76].

LasB, also known as pseudolysine, is one of the best-characterized proteases of *P. aeruginosa*. This 33 kDa enzyme is Ca^2+^ and Zn^2+^ dependent and belongs to the thermolysin-type M4 peptidases family [77]. Its expression is regulated by the master regulator LasR of the quorum sensing system [78]. It has been reported that, in vitro, LasB cleaves C1q and C3, both in the soluble phase and bound to bacterial surfaces [79,80,81]. In addition, LasB inactivates the purified components C5a [82], C5, C8, and C9 [79].

The alkaline protease AprA, also known as aeruginolysin, is a metalloendopeptidase of about 50 kDa [that belongs to the B subfamily of the M10 peptidase family [83]. As LasB, AprA is regulated by the master regulator LasR and is Zn^2+^ dependent [84]. In vitro experiments have demonstrated that AprA degrades the C2 component in vitro, inhibiting the activation of the CP and LP [85]. Degradation of the second complement component by AprA inhibits C3b formation and, subsequently, reduces bacterial opsonization, C5a formation, and neutrophil recruitment, allowing the bacteria to elude phagocytes [85]. Moreover, AprA cleaves C1q [80,85], C3 [81,85], and C5a [82].

Despite the fact that several in vitro studies have demonstrated that both purified proteases cleave several components of the immune system, their contribution to *P. aeruginosa* infections in vivo remain poorly investigated. Using a murine model of pneumonia, it has been demonstrated that LasB is important for the establishment of *P. aeruginosa* respiratory infection [86,87]. However, little is known about the role of LasB or ApraA in *P.aeruginosa* bloodstream infections. In an early study, Wretlind et al., showed that in experimental bacteremia in mice, LasB did not promote *P. aeruginosa* virulence [88]. More recently, we have demonstrated that the ability of the isogenic deficient-AprA or LasB deficient mutants to survive in blood and to cause bacteremia in a murine model of systemic infection is similar to the parental strain [81]. This is probably a consequence of the redundancy of the multiple virulence factors produced by this pathogen to resist complement. According to our results, it is likely that both exoproteases contribute to the virulence of *P. aeruginosa* once the infection is already established, but not in the early steps when the innate immune system is faster than the synthesis of both proteases by the microorganism.

Another exoprotease of *P. aeruginosa* of particular interest is the protease IV or lysyl endopeptidase (PIV or PrpL). The production of this 26 kDa serine protease is relatively conserved among *P. aeruginosa* strains, suggesting an essential role for this protein [89]. As occurs with the other exoproteases, PIV is regulated by the quorum system, but above all, it is controlled by iron availability. PIV is involved in the degradation of components of the complement system such as C3 and C1q [90].

Finally, the Pseudomonas Small Protease (PASP) is an 18.5 kDa leucine that is aminopeptidase Zn^2+^ dependent. It is involved in the degradation of structural components such as type I and IV collagen [91], the alpha and beta chains of fibrinogen [92], LL-37 [92], and the alpha chain of the C3 complement component [92].

#### 4.3.2. Ecotin

Another inactivator of the complement proteins expressed by *P. aeruginosa* is Ecotin. This protein is a periplasmatic and a matrix protein found in biofilm that prevents activation of the LP by inactivating MASP-1 and MASP-2, as well as the AP by inhibiting MASP-3, a Factor D activator component [93].

#### 4.3.3. Lytic Polysaccharide Monooxygenase CbpD

Lytic polysaccharide monooxygenases cleave polysaccharides by oxidation and are associated with bacterial virulence. It has been recently shown the *P. aeruginosa* CbpD function is essential for the virulence of the microorganism. Askarian et al., observed that the catalytic activity of this enzyme impairs C5 convertase assembly, resulting in the attenuation of the terminal pathway of the complement [94]. Thus, CpD-deficiency reduced the virulence of *P. aeruginosa* in a murine model of systemic infection.

## 5. Conclusions

Clinical and experimental evidence has demonstrated that the complement system is an essential effector of the immune system in *P. aeruginosa* bloodstream infections. Over the last decades, different strategies used by this pathogen to counteract its effects have been identified. While it seems clear that in this pathogen the expression of the LPS O side chain is the main strategy to resist complement in blood, the other mechanisms are apparently redundant, but they ensure the viability of the microorganism, and their contribution is probably essential, depending on the entry route to the bloodstream. Thus, the complement-resistant phenotype may result from the combination of multiple virulence factors expressed by each isolate which, eventually, may be expressed transiently by a subpopulation of cells to evade the immune system and persist in the blood.

Future studies should be focused, on one side, on the host, and use global approaches to identify novel serum proteins that interact with *P. aeruginosa* and characterize the consequences of these interactions on complement activation and deposition in the bacterial cell. After all, the outcome of *P. aeruginosa* bloodstream infections will depend on the virulence factors expressed by the bacteria and the multiple interactions between the host proteins and bacteria. One example are the complement factor H related proteins that are recruited by different pathogens [95], including *P. aeruginosa* [68,69], but their biological significance is still poorly understood. On the other side, the characterization of the strategies displayed by *P. aeruginosa* to counteract the complement attack opens up the development of novel tools to treat bloodstream infections. The use of bacteriophages against the bacterial structures involved in complement resistance are promising candidates [96,97].

## Figures and Tables

**Figure 1 microorganisms-11-00664-f001:**
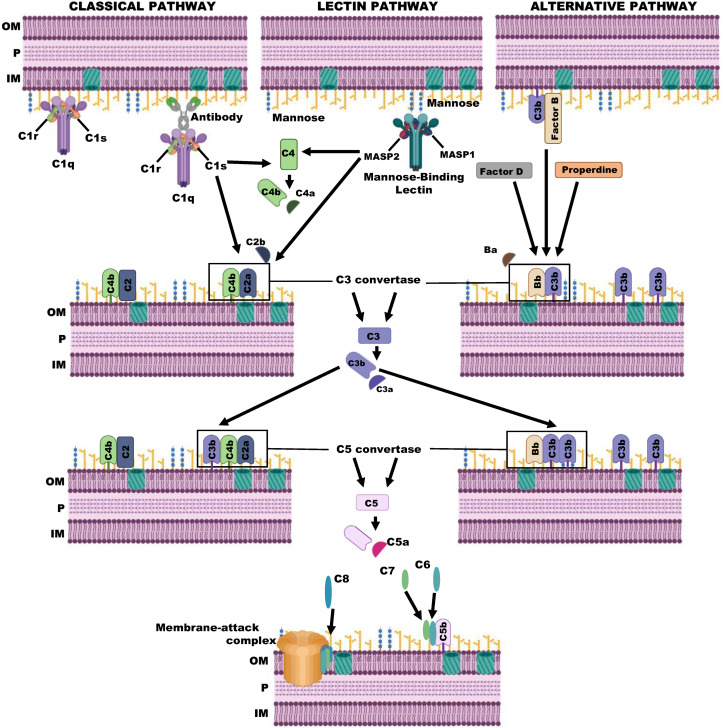
Schematic representation of the complement system activation on a bacterial cell. Details are explained in the text. (Created with BioRender.com, accessed on 31 January 2023)**.** OM: Outer membrane; P: Periplasm/peptidoglycan; IM: Inner membrane.

**Figure 2 microorganisms-11-00664-f002:**
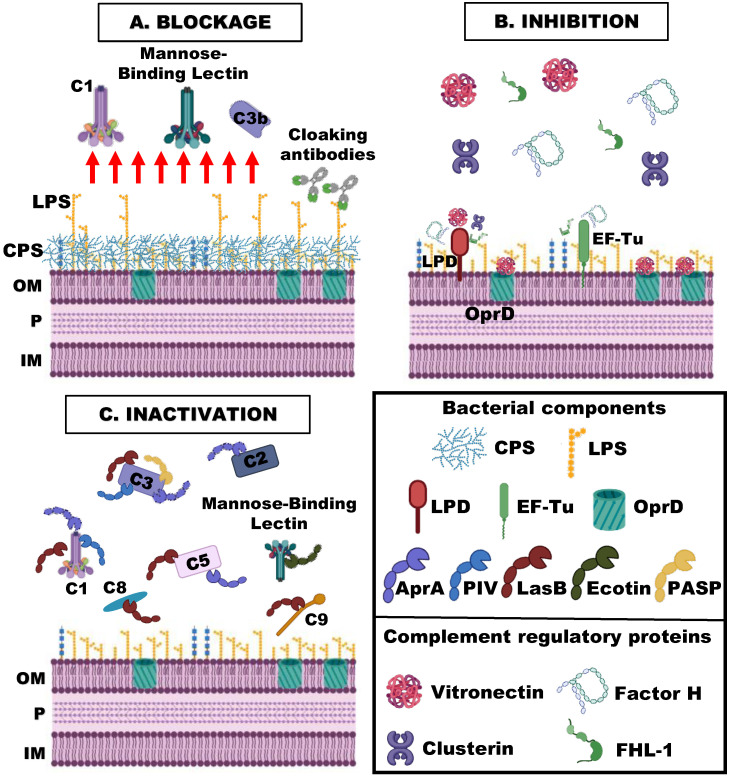
Diagram of the general strategies exploited by *P. aeruginosa* to elude the complement system attack. (Created with BioRender.com). (**A**) Blockage of the interaction of the complement components with the bacterial cell; (**B**) the binding of regulatory complement regulatory proteins that inhibit complement activation; and (**C**) the inactivation of complement proteins. OM: Outer membrane; P: Periplasm/peptidoglycan; IM: Inner membrane; CPS: Capsular polysaccharides; LPS: Lipopolisaccharide; LPD; Dihydrolipoamide dehydrogenase; EF-Tu: Elongation Factor Tu; OprD: Outer membrane protein OprD; AprA: Alkaline protease A; PIV: Protease IV; LasB: Elastase B; PASP; Pseudomonas Small Protease.

**Figure 3 microorganisms-11-00664-f003:**
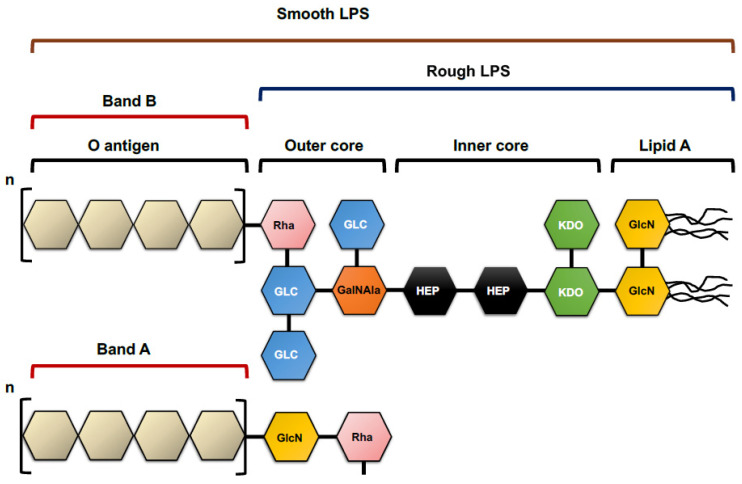
Structure of the *P. aeruginosa* LPS. Monosaccharides depicted are hexagons of different colors. Details are explained in the text. GlcN: Glucosamine; KDO: D-manno-oct-2-ulosonic acid; HEP: Heptose; GLC: Glucose; Rha: Rhamnose; GalNAla: Galactosamine.

## Data Availability

Not applicable.

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
