# Peer review of "Pseudomonas aeruginosa and the Complement System: A Review of the Evasion Strategies"

_microorganisms, 2023, doi:10.3390/microorganisms11030664_

Round 1

Reviewer 1 Report

The review by González-Alsina and colleagues is overall well-written and interesting. After an initial description of the activation pathways of the complement cascade, the authors analyze the main factors that P. aeruginosa could exploit in bloodstream infections to counteract the complement action.
The major concern is about the figures. They should be redone entirely, less confusing, and of better quality. In particular, some symbols used could have a better resolution. The caption of the figures should be detailed to facilitate the understanding of the images.

Minor points
Lines 34-35: rewrite the sentence in a more straightforward way
Lines 58-59: you should detail the figure caption or, in the alternative, add the sentence "Details are explained in the text."
Lines 62-66: sentence too long and unclear
Line 182: delete "model."
Lines 246-281: too long, rewrite, avoiding unnecessary details.
Lines 303-304: a reference is needed.
Lines 374-377 and 383-385: references are needed

Reviewer 2 Report

This review paper by Gonzalez-Alsina et al. reports on the mechanisms used by the bacterial pathogen Pseudomonas aeruginosa (Pa) to resist the complement system, which is one of the main components of innate immunity in mammals.

I understand this review is part of a Special Issue of <Microorganisms> on the subject "Bacterial Pathogens Associated with Bacteremia".

The review is well organized and, overall, it is well written.  It will require some revision, however, in a few places were word usage and grammar could be improved.  For instance (just as a couple of examples) in line 32 it would be best to use the following wording: '...has made the World Health Organization include Pa in its list of most dangerous pathogens [11].'  In line 47, the word "lysis" (which is not a verb) should be replaced with the word 'lyse', and in line 48, the wording "...the phagocyte cells...", could be simply replaced by the word 'phagocytes'.

I believe the review provides a sound contribution to the field covered in the Special Issue, given the authors address the following specific comments:

ABSTRACT: Somewhere in the Abstract the authors should mention that the focus of the review will be on Pa bacteremia and systemic infections.  This is necessary to justify the inclusion of this review as part of the Special Issue on "Bacterial Pathogens Associated with Bacteriemia".

INTRODUCTION and lines 34-37.  In the lines mentioned, the concept should be introduced that in that timeline (only a few hours) there is no time to mount an adaptive immune response, and therefore, it is up to the innate immune response to counteract the infection before death happens.  In this respect, the authors also should provide in the INTRODUCTION a comprehensive explanation of the different role that the complement system might play during systemic versus lung infections caused by Pa.  It is obvious that during systemic infections, Pa would encounter elements of the complement system, and that this system should provide a strong defense against such infections. On the other hand, authors should mention what is the role of complement during lung infections, and how likely it is for Pa to encounter complement components during its establishment in the lung mucosa (even in biofilms).

Line 43 = Section 2. For the readers' benefit and to let them know which of the references cited are, in turn, review papers,  the authors should provide a list of the reviews cited, stating that the literature on the complement system is overly abundant and that the reader is referred to recent reviews to expand knowledge on this subject.

Line 187. "...type of infection..."  The difference in the type of infection treated in these two studies should be discussed in more detail.  One paper deals with systemic spread (thereby involving bacteremia) and the other with lung infection (which does not necessarily involve bacteremia).  In my view there is no contradiction between these studies.  They simply indicate that  that while the LP seems to be important in controlling bacteremia, it is not that important in lung infections, which do not require the pathogen to enter the bloodstream.

Section 4.1.1. Lypopolysaccharide.  Since LPS has been established as a complement-binding molecule in Section 3, the title of section 4.1.1. should be changed to reflect the fact that the blockage of complement components is based on structural/chemical modifications to the LPS, and not to the native molecule itself.  Also, somewhere in this section it should be stated that a long O-chain (leading to a smooth cell surface phenotype) is NOT a complement-binding structure.

Line 326.  Please change the word "outcomes" to 'results'.  What does C2 and C3 (mentioned in this line) stand for?  I assume that they are not referring to the complement components, but without a definition it is very confusing.

Section 4.1.3.  Is there any evidence that Pa forms biofilms during systemic infections or during bacteremia?  It is important to mention this to contrast systemic versus lung infections.  If biofilm formation is not a recognized Pa resistance mechanism during systemic infections, then this should be clearly stated here.  I believe that since the review is (or should be) focused on bacteremia and systemic infections, the discussion of lung biofilms should be placed in its right perspective.

FIGURES.  The "Bacterial cell" label used in Figs 1 and 2 is a bit misleading, since only a section of the bacterial cell wall is shown in the diagrams.  Authors should label the layers depicted as 'Outer membrane', 'Periplasm/Peptidoglycan', and 'Inner membrane'.  The legends of the figures should include an explanation of the abbreviations/labels used in the diagrams.  That is what is the meaning of C1r, C1q, MASP, EF-tu, FHL-1, etc., etc.).  Also, the legend of Fig. 3 should include an explanation of what the hexagons (and their colors) represent in the diagrams.

Reviewer 3 Report

Regarding the Manuscript ID microorganisms-2220804 Title “Pseudomonas aeruginosa and the complement system: A review of the evasion strategies.”

Even the information is clear and write well, the authors use a lot of old references, more than 30% of them are 22 years old, 33% are 20 years old and only 14% are between 5 years published.

Most of the reviews in any journal has references of no more than 5 years old, then is an invitation to focus the information on the last 5 years. I am sure that many of the information are already in scholar books and/or part of other reviews.

The topic is important but the authors need to focus on last references to support the manuscript with the last knowledge and propose new hypothesis on the role of the complement and how can be fight the different virulence factors of Pseudomonas aeruginosa

Round 2

Reviewer 1 Report

The authors have addressed all my issues. I agree to publication in the revised form.

Author Response

Thank you for your revision.

Reviewer 3 Report

Regarding the new version of this manuscript, still there are very old references and I found that some of them are included in other recent refences. Like

 Larsen, G. L.; Mitchell, B. C.; Harper, T. B.; Henson, P. M. The pulmonary response of C5 sufficient and deficient mice to 680 Pseudomonas aeruginosa. Am. Rev. Respir. Dis. 1982, 126 (2),, 306–311. Reference cited in Mishra et al 2015

Mueller-Ortiz, S. L.; Drouin, S. M.; Wetsel, R. A. The alternative activation pathway and complement component C3 are critical 682 for a protective immune response against Pseudomonas aeruginosa in a murine model of pneumonia. Infect. Immun. 2004, 72 (5),, 683 2899–2906. Reference cited in Mishra et al 2015

Albertí, S.; Alvarez, D.; Merino, S.; Casado, M. T.; Vivanco, F.; Tomás, J. M.; Benedí, V. J. Analysis of complement C3 deposition 708 and degradation on Klebsiella pneumoniae. Infect. Immun. 1996, 64 (11),, 4726–4732. Reference cited in Mishra et al 2015

Jensen, E. T.; Kharazmi, A.; Garred, P.; Kronborg, G.; Fomsgaard, A.; Mollnes, T. E.; Høiby, N. Complement activation by 711 Pseudomonas aeruginosa biofilms. Microb. Pathog. 1993, 15 (5),, 377–388. Reference cited en Donland and Costerton. 2002 and in Qadi et al. 2017

Costerton, J. W.; Lewandowski, Z.; Caldwell, D. E.; Korber, D. R.; Lappin-Scott, H. M. Microbial biofilms. Annu. Rev. Microbiol. 804 1995, 49, 711–745. Reference cited en Donland and Costerton. 2002

Suggestion: 

The reference “Cerquetti, M. C.; Sordelli, D. O.; Bellanti, J. A.; Hooke, A. M. Lung defenses against Pseudomonas aeruginosa in C5-deficient mice 678 with different genetic backgrounds. Infect. Immun. 1986, 52 (3) 853–857.” Could be usd a more actual reference that include Cerquetti and talk about the same issue, like “Wilson et al., 2007. Microbiology. 153:968-979. doi: 10.1099/mic.0.2006/002261-0”

Find the way to use recent references to citate old references

Author Response

Thank you for your suggestions. However, in the benefit of the readers, and in our opinion, references should be the original works describing the results indicated in the manuscript rather than more recent publications that only reference the original work but do not describe in detail the research. Therefore, old references (but original) should not be replaced by more recent references.

In addition, references should give credit to the research groups that provided the original results. For these reasons, we think that the old references included in more recent references shouldn't be deleted.